# Identifying Gut Microbiota Conditions Associated with Disease in the African Continent: A Scoping Review Protocol

**DOI:** 10.3390/mps6010002

**Published:** 2022-12-24

**Authors:** Sara M. Pheeha, Jacques L. Tamuzi, Samuel Manda, Peter S. Nyasulu

**Affiliations:** 1Division of Epidemiology and Biostatistics, Faculty of Medicine and Health Sciences, Stellenbosch University, Cape Town 7500, South Africa; 2Department of Chemical Pathology, Faculty of Medicine and Health Sciences, Sefako Makgatho Health Sciences University, Pretoria 0208, South Africa; 3National Health Laboratory Health Service, Dr. George Mukhari Academic Hospital, Pretoria 0208, South Africa; 4Department of Statistics, Faculty of Natural and Agricultural Sciences, University of Pretoria, Pretoria 0028, South Africa

**Keywords:** gut microbiota, gut microbiome, human health, diseases, dysbiosis, eubiosis, F/B ratio, gut diversity/richness, taxonomic profiles, Africa

## Abstract

The gut microbiota has been immensely studied over the past years because of its involvement in the pathogenesis of numerous diseases. However, gut microbiota data in Africa are limited. Therefore, it is crucial to have studies that reflect various populations in order to fully capture global microbial diversity. In the proposed scoping review, we will describe the gut microbiota’s appearance in terms of gut microbiota markers, in both health and disease in African populations. Relevant publications will be searched for in the PubMed, Scopus, Web of Science, Academic Search Premier, Africa-Wide Information, African journals online, CINAHL, and EBSCOhost and Embase databases. We will focus on articles published between January 2005 and March 2023. We will also determine if the studies to be included in the review would provide enough data to identify quantifiable gut microbiome traits that could be used as health or disease markers, identify the types of diseases that were mostly focused on in relation to gut microbiota research in Africa, as well as to discover and analyze knowledge gaps in the gut microbiota research field in the continent. We will include studies involving African countries regardless of race, gender, age, health status, disease type, study design, or care setting. Two reviewers will conduct a literature search and screen the titles/abstracts against the eligibility criteria. The reviewers will subsequently screen full-text articles and identify studies that meet the inclusion criteria. This will be followed by charting the data using a charting tool and analysis of the evidence. The proposed scoping review will follow a qualitative approach such that a narrative summary will accompany the tabulated/graphical results which will describe how the results relate to the review objectives and questions. As a result, this review may play a significant role in the identification of microbiota-related adjunctive therapies in the African region where multiple comorbidities coexist. Scoping review registration: Open Science Framework.

## 1. Introduction

The gut microbiota has attracted much research over the past years and is believed to be involved in the pathogenesis and progression of numerous diseases [1,2]. The gut microbiota is a complex, dynamic, and spatially diverse ecosystem that consists of countless microorganisms (bacteria, fungi, archaea, and viruses) that interact with each other, and with the human host in the gastrointestinal ecosystem [3]. It is an immense microbial community that plays a great role in maintaining human life and is therefore considered to be the “essential organ” of the human body [3] or the additional “endocrine organ” [4].

Under normal conditions, the gut microbiota is responsible for several core functions to benefit the human host [5]. These include metabolizing proteins and complex carbohydrates [6], production of hormones and neurotransmitters [7], biosynthesis of certain vitamins and essential amino acids, short-chain fatty acids production [8,9] and the synthesis of particular lipopolysaccharides [10]. The gut microbiota also interacts with the host’s innate and adaptive immune system to maintain intestinal homeostasis and inhibit inflammation [11].

Despite the lack of a “gold standard” reference for the composition of human gut microbiota, as it is different for everyone [10,11], there are certain gut microbiota conditions that are more favourable to the human host as compared to others. The term used to describe a well-balanced and ideal gut environment is termed “eubiosis” [12], while the opposite is known as” dysbiosis”. The latter is described as reduced diversity of the gut microbiota [13], which results in an imbalance in the composition and metabolic roles of the microbiota [14]. Dysbiosis can either be a result of disease or lead to the development of disease [15].

Illnesses such as metabolic disorders, cardiovascular and cerebrovascular diseases, autoimmune disorders, inflammatory bowel disease [16], psychotic disorders [17], and cancer, could result from an imbalanced gut microbiota [18,19]. Human immunodeficiency virus (HIV) and Tuberculosis (TB) have also been associated with gut dysbiosis. Patients who are HIV positive present with reduced alpha-diversity and increased Enterobacteraceiae [20]. Moreover, patients with TB who are taking anti-TB drugs experience dysbiosis as a result of the treatment [21].

An impaired interaction between gut microbiota and the mucosal immune system, as seen in dysbiosis can also lead to an increased abundance of potentially pathogenic gram-negative bacteria and their associated metabolic changes, therefore disrupting the epithelial barrier [22] and increasing susceptibility to infections [6].

There is extensive research that is available regarding the characterization of gut microbiota in both health and disease in Western populations [1]. These studies have generally found that the composition of gut microbiota differs significantly between healthy individuals and those who are diseased [23]. They have also reported that some of the individuals who have diseases such as cardio-metabolic, irritable bowel, and autoimmune disorders present with high abundances of bacterial pathogens which include *E. coli*, *S. aureus*, and *C. difficile* [23]. 

Gut microbiota data are limited in Africa, and the effects of microbial diversity on health and disease in the continent are not well understood. Therefore, studies that elucidate different gut microbiota states in various conditions in Africans are required. 

Our focus on African individuals is justified by the fact that people living in Western and African countries are exposed to unique environments and follow different diet, sanitation, and hygiene practices [20], all of which can modify gut microbiota. Accordingly, prospective gut microbiota studies must prioritize studies that reflect various populations in order to fully capture global microbial diversity [20].

Moreover, Sub-Saharan Africa in particular has the highest burden of HIV/AIDS [24]. The gut microbiota has been linked to HIV-infected patients on antiretroviral therapy (ART). According to the evidence, patients on ART treatment who have poor CD4+ T-cell recovery have higher levels of microbial translocation and immune activation [25]. A recent study conducted in Zimbabwe demonstrated that HIV-infected children have altered gut microbiota, and that prolonged ART use may restore their microbiota richness [26]. Africa is also among the regions with a high prevalence of tuberculosis (TB). People with a latent TB infection may either completely clear their infection or develop active TB disease depending on their immunological health, which can be impacted by the gut microbiota [21]. Furthermore, the host microbiota contributes to an early protection against Mycobacterium TB colonization of the human lung [21]. 

Lastly, African countries like South Africa are currently experiencing a surge in the number of metabolic conditions such as type 2 diabetes mellitus (T2DM) and obesity because of the adoption of Western diets and decreased levels of physical activity [27]. Both these conditions are believed to affect gut microbiota, as they can cause gut dysbiosis which may lead to the development of co-morbidities that are associated with disease progression [20]. 

A preliminary search of PubMed, the Cochrane Database of Systematic Reviews and Joanna Briggs Institute (JBI) Evidence Synthesis was conducted and one survey on a similar topic was conducted by Brewster et al., 2019 [20]. The review specifically looked at Africa; however, it lacked a clear methodology. The search methodology was not described and there were no explicit and clear criteria for selection of articles. A systematic review conducted by Allali et al., 2021 [1] was also identified which also surveyed Africa as a whole. They, however, looked at the entire human microbiome, with gut microbiota being a part of their focus. Their methodology was clear and well-described; however, their limitation is that they considered studies that only employed new-generation sequencing as the technology to analyze the microbiome. Due to this restriction, they might have missed other important studies that used alternative technologies. 

The main objective of the proposed scoping review is to map out relevant literature related to gut microbiota and to further describe the gut microbiota’s appearance in terms of gut microbiota markers such as Firmicutes/Bacteroidetes (F/B) ratio, gut diversity, gut richness, and taxonomic profiles, in both health and various disorders in African populations. Our secondary objectives are to; (i) determine if the studies to be included in the review would provide enough data to identify quantifiable gut microbiome traits that could be used as health or disease markers, (ii) identify the types of diseases that were mostly focused on in relation to gut microbiota research in Africa (i.e., diseases of public health significance), and (iii) to ultimately discover and analyze knowledge gaps in the gut microbiota research field in the continent.

The proposed scoping review aims to comprehensively review gut microbiota research (regardless of the type of technology used to analyze gut microbiota) conducted in Africans, and identify gut microbiota conditions that are associated with health and various diseases.

## 2. Materials and Methods

The review will be conducted in accordance with the JBI methodology for scoping reviews [28], as well as the Preferred Reporting Items for Systematic Reviews and Meta-Analysis extension for scoping reviews (PRISMA-ScR) [29]. In the case of necessary amendments, the authors will ensure that the JBI methodology for scoping reviews and PRISMA-ScR guidelines remain adhered to. The amendments will be transparently reported. 

### 2.1. Scoping Review Questions

The scoping review will answer the following questions:How does the gut microbiota differ between health and various disease states in African individuals in terms of gut microbiota markers such as F/B ratio, gut diversity, gut richness, and taxonomic profiles?Are the studies included in the scoping review sufficient to provide enough data to identify quantifiable gut microbiome traits that could be used as health or disease markers?Which conditions were mostly covered by the existing gut microbiota research in Africa, and what are the current knowledge gaps in gut microbiota research in the continent?

### 2.2. Eligibility Criteria and Study Types

The eligibility criteria for the scoping review were established based on the PCC framework as follows:

Participants: The proposed scoping review will include studies involving African countries regardless of race, gender, age, health status, disease type, study design, or care setting. Studies that did not include any human participants will be excluded from the review.

Concept: The review will include studies that looked at different disorders, provided that the gut microbiota was described and characterized in that condition. Articles examining the gut microbiota in apparently healthy individuals will also be considered for the proposed review. We will also include articles that used any type of technology to profile gut bacteria identified from human samples. Additionally, outcomes relating to gut microbiota such as F/B ratio, gut diversity, gut richness, taxonomic profiles, and others will be identified and recorded as part of the scoping study. Studies that did not describe and characterize the gut microbiota will be excluded.

Context: The review will consider studies that only included African individuals. Studies that were not conducted on African individuals or those that included a mixture of African participants with participants from other non- African countries will be excluded from the review.

The review will consider both experimental and quasi-experimental study designs including randomized controlled trials, non-randomized controlled trials, before and after studies, and interrupted time-series studies. In addition, analytical observational studies including prospective and retrospective cohort studies, case-control studies, and analytical cross-sectional studies will be included. Reviews of all types along with editorials, opinion papers, commentaries, news, and notes will be excluded from the scoping study.

### 2.3. Search Strategy

The search strategy will aim to locate both published and unpublished studies. An initial limited search of PubMed was undertaken to identify articles on the topic. The text words contained in the titles and abstracts of relevant articles, and the MeSH terms used to describe the articles were used to develop a full search strategy for PubMed (see Appendix A). The search strategy, including all identified keywords and Medical Subject Headings-MeSH terms, will be adapted for each included database or information source. Our faculty librarian was consulted regarding developing a precise search strategy. The reference list of all included sources of evidence will be screened for additional studies. 

Studies published only in English will be included because we would like to save human and financial resources, since it is extremely challenging to identify suitable people to assist with the translation of articles written in alternative languages to English. Moreover, in the case where someone is identified it usually takes long periods of time to finally receive the translated paper/s. The proposed scoping review will include all studies that describe and characterize gut microbiota in relation to health and various disease states that were published from as early as January 2005 until March 2023. This is because as it is, the gut microbiota in Africa as a whole is understudied; therefore, it is inevitable that only a small number of studies will be available as compared to Western countries. 

The databases to be searched include PubMed, Scopus, Web of Science, Academic Search Premier, Africa-Wide Information, African journals online, CINAHL, EBSCOhost and Embase. Sources of unpublished studies/gray literature to be searched include government reports, policy statements, and issues papers, conference proceedings, pre-prints and post-prints of articles, theses, and dissertations, as well as research reports. The Cochrane Methodology Register (CMR) and World Health Organization International Clinical Trials Registry platform will also be searched to locate unpublished studies. Authors who wrote relevant reports (e.g., conference proceedings, theses, dissertations etc.) will be contacted to find out if they have applicable research studies that have not been published.

### 2.4. Selection of Studies

Following the search, all identified citations will be collated and uploaded into EndNote v.X20 (Clarivate Analytics, PA, USA) and the duplicates as well as reviews, editorials, and opinion papers will be removed by two reviewers separately. Following a pilot search, titles and abstracts will then be screened by two independent reviewers for assessment against the eligibility criteria for the review. Once the duplicates are removed, two reviewers will independently screen the title and abstract of the remaining articles against the inclusion and exclusion criteria. The full texts of the studies that passed this stage will be retrieved and further independently reviewed by two independent reviewers based on the eligibility criteria. 

The reviewers will then compare their results, and if any disagreements arise at any of the stages of the process, a third reviewer will be consulted. Studies that do not meet the inclusion criteria will be documented and reasons for exclusion will be provided in the scoping review. The results of the search and the study inclusion process will be reported in full in the final scoping review and presented in a PRISMA flow diagram [30].

### 2.5. Charting the Data

Data will be extracted from papers included in the scoping review by two reviewers using a charting tool developed by the reviewers. The data extracted will include the first author’s name and year of publication, description of gut microbiota, disease under study and whether it is of public health significance or not, country of origin, study setting (urban or rural), the technologies used to analyze the gut microbiota, study design, sample size, type of sample used, demographic characteristics (age, gender, race), the study aims and objectives as well as general key findings of the studies. 

A draft charting form is provided (see Appendix B). The draft charting tool will be modified and revised accordingly during the process of extracting data from each included study. Modifications will be detailed in the scoping review. Any disagreements that arise between the reviewers will be resolved through discussion or with additional reviewers. If appropriate, authors of papers will be contacted to request missing or additional data, where required.

### 2.6. Analysis of Data and Presentation of Results

The Joanna Briggs Institute (JBI) methodology will be used to develop this scoping review protocol [31]. The PRISMA for scoping reviews (PRISMA-ScR) checklist [29] will be used to guide our study’s findings reporting. The proposed scoping review will follow a qualitative approach and the findings will be presented in tables, graphs, maps and diagrams. A narrative summary will accompany the tabulated/graphical results and will describe how the results relate to the review objective and questions. The narrative synthesis will describe relevant data about gut microbiota diversity, gut microbiota richness, gut microbiota diversity, taxonomic profile, F/B ratio, and other gut microbiota measurements in different conditions, and will also include all phyla, families, genera, and species of gut bacteria that were identified in both diseased and non-diseased participants. Because the purpose of a scoping review is to aggregate evidence and present a summary of the evidence rather than to evaluate the quality of the individual evidence, no formal assessment of the bias assessment will be included in this review.

## 3. Expected Outcomes and Implications

The gut microbiome is crucial for overall health. Life threatening diseases such as HIV, Diabetes, Cardiovascular and Cerebrovascular conditions may be influenced by an imbalance of hazardous and beneficial microorganisms in the gut. Moreover, the gut microbiota of an individual may affect their propensity to contract infectious diseases and play a role in chronic gastrointestinal conditions including Crohn’s disease and irritable bowel syndrome.

Given that the African region has one of the highest burdens of HIV, and other diseases such as tuberculosis, arterial hypertension, and T2DM, this review could be useful in understanding the gut microbiota’s role in disease development and pathogenesis.

The proposed review is also relevant in Africa, where the new concept of “Host Microbiota Directed-Therapies” as potential adjunctive strategies to improve treatment efficacy, reduce treatment duration, and/or prevent relapses are being explored [21]. Other important review fields will include mapping geographical location and the impact of various treatments on gut microbiota, such as Antiretrovirals, TB regimens, hypertensive and diabetes mellitus drugs.

In the proposed scoping review, we will synthesize emerging evidence on gut microbiota that may be applicable in evidence-based practices in Africa. 

## 4. Limitations

The lack of critical appraisal in the proposed scoping review, on the other hand, may weaken our findings. Unpublished studies may help to minimize bias brought on by selective publication, boost the effectiveness of systematic reviews, and lessen research waste. However, we may not be able to completely avoid research waste due to the challenges associated with locating relevant unpublished research studies. Moreover, unpublished studies may also lower the methodological quality of the review than those that are peer reviewed and published. 

## Data Availability

Not applicable.

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
