# Peer review of "Identifying Gut Microbiota Conditions Associated with Disease in the African Continent: A Scoping Review Protocol"

_mps, 2022, doi:10.3390/mps6010002_

Round 1

Reviewer 1 Report

The authors presents a scoping review protocol to identifye gut microbiota conditions associated with disease in Africans

The methods are consistent with JBI methodology for scoping reviews as well for PRISMA-ScR, and are well explained.

The search strategy will aim to locate both published and unpublished studies. It is unclear how the unpublished studies, mostly uncurated, can be merged and stand in the same position as published ones.

The results presented are poor, only a Table A1, a search strategy in PubMed, which means only published results.

As the authors point up in the introduction one survey on a similar topic was conducted in 2021 in a systematic review with the entire human microbiome. The diference with the present work is that this one focuses only in gut microbiota and that it takes in account not only next generation sequencing but other technologies. It will be good to know if the search of both strategies gives significantly different results.

Reviewer 2 Report

The manuscript entitles “Identifying gut microbiota conditions associated with disease in Africans: a scoping review protocol” (mps-2068516) has been presented to “Methods and protocols” in the Section “Public Health Research”, and in the type “Protocol”.

This review study is based on the gut microbiota data in Africa is limited and for this reason it is important to have studies that reflect various populations to fully capture global microbial diversity in terms of gut microbiota markers in this population.

In the summary it is advisable to include the period that includes the remission, this allows it to relate to other works on this subject. Likewise, the databases that have been used to identify the articles and the number of articles that have finally been studied must be included.

keywords should appear review.

The introduction lays the foundations for the importance of the microbiota today. The objectives set out in the introduction line 70 to 89 I suggest that they be synthesized. The objective is better than this at the end of the introduction and connects with the proposed objective of line 114 - 116. The introduction should put more information in relation to the diseases with which HIV, tuberculosis is related…. explaining the importance of the microbiota in these diseases.

Materials and methods.

The type of review must be included.

JBI the first time it is used the meaning must be indicated (Joanna Briggs Institute). The same occurs with other acronyms that appear in the text, the first time they appear they must be informative.

The date that corresponds to the review carried out must be indicated, both the period to which the articles correspond and the period in which the review was carried out to connect with other works.

I don't understand the term unpublished study from line 166 in a review please explain.

Reviewer 3 Report

Reviewer comments and suggestions

The authors in this review discussed the gut microbiota's appearance in terms of gut microbiota markers, in both health and disease in African populations. The studies involving African countries regardless of race, gender, age, health status, disease type, study design, or care setting were taken into account. Two reviewers will conduct a literature search and screen the titles/abstracts against the eligibility criteria. The reviewers will subsequently screen full-text articles and identify studies that meet the inclusion criteria. Hence the authors suggested that this review may play a significant role in the identification of microbiota-related adjunctive therapies in the African region where multiple comorbidities coexist. 

The paper was well-written, and a few minor modifications are needed to work on the manuscript.

  1. Line 43-44 and similar comments of 56-57 One reference was not enough, please mention few
  2. Comments for line 62 The author mentions extensive research and adds only one reference that was already used
  3. Line 63-64 It would be nice if the authors plan to discuss more here
  4. Line 86-87 why suddenly does the para need to be discussed HIV not with any other diseases, is there was higher prevalence than others
  5. Section 2.6 It should be more discussed
  6. Line 235-236 Here the author has to mention why the microbiota study was important to discuss
  7. Line 244 if possible the authors could mention a few lines of limitations 

Round 2

Reviewer 2 Report

In relation to the article, I have carefully reviewed the new version of the manuscript entitled “Identifying gut microbiota conditions associated with disease in Africans: a scoping review protocol” (mps-2068516) has been presented to “Methods and protocols” in the Section “Public Health Research”, and in the type “Protocol”.

In the first place, I would like to state the importance of the summary for the readers of the magazine, since the interest in reading the complete article is decided based on the information contained. Therefore, the information that it must contain must be as accurate as possible.

I am sorry for the mistake in the word " remission" since it should be a review, many reviews are currently being carried out, therefore it is essential to identify what period of time the reviews included in this protocol comprise. It is also essential to indicate the databases to be reviewed in the summary, not only in the material and methods section. On the other hand, it is also important to indicate what type of review is going to be carried out (qualitative, quantitative, systematic, meta-analysis...) although all these sections must also be indicated in material and methods.

The authors have not included review keywords among the keywords.

Unpublished articles are difficult to assess in terms of their scientific evidence.
